# Migraine Pathophysiology Revisited: Proposal of a New Molecular Theory of Migraine Pathophysiology and Headache Diagnostic Criteria

**DOI:** 10.3390/ijms232113002

**Published:** 2022-10-27

**Authors:** Yasushi Shibata

**Affiliations:** Mito Medical Center, Mito Kyodo General Hospital, University of Tsukuba, Tsukuba 310-0015, Ibaraki, Japan; yshibata@md.tsukuba.ac.jp; Tel.: +81-29231-2371

**Keywords:** migraine, CGRP, trigeminal nerve, cortical spreading depression, hypothalamus

## Abstract

Various explanations for the pathophysiology of migraines have been proposed; however, none of these provide a complete explanation. The author critically reviews previous theories and proposes a new molecular theory of migraine pathophysiology. The diagnosis of primary headaches is generally based on clinical histories and symptoms only because there is no reliable diagnostic examination. The author proposes a new classification system and set of diagnostic criteria for headaches based on molecular markers.

## 1. Introduction

Various explanations for the pathophysiology of migraines have been proposed, including vascular theory, neural theory, and trigeminovascular theory [1,2,3]. However, none of these provide a complete explanation, and there are considerable contradictory data. This lack of a comprehensive understanding of migraine pathophysiology is a major obstacle to the advancement of migraine research and treatment.

Recent research indicates that calcitonin-gene-related peptides (CGRPs) play an important role in the mechanism of migraine [4,5,6]. A study has also found that 5-HT_1F_ receptor agonists are clinically efficacious in the treatment of migraine attacks [7]. However, the underlying molecular mechanisms of migraine have not been clearly presented. In this article, the author critically reviews previous theories and proposes a new molecular theory of migraine pathophysiology.

There is currently an international classification system for headache disorders, and this is used as the world standard for headache diagnoses and classification [8]. The diagnosis of primary headaches, such as migraine, tension-type headaches (TTH) and cluster headaches, is generally based on clinical histories and symptoms only because there is no reliable diagnostic examination. More objective physiological measures for migraine diagnosis cannot be developed until we fully understand the pathophysiology of migraine. In this article, the author proposes a new classification system and set of diagnostic criteria for headaches based on molecular markers. These pathophysiological diagnostic criteria are easy to understand and are clinically valuable, allowing for more accurate diagnoses and treatment choices.

## 2. The Vascular and Trigeminovascular Theories of Migraine Pathophysiology

Because migraines have pulsatile properties, it has been suggested that they may have vascular origins. On this basis, the first theory of the pathophysiology of migraines was the vascular theory of Graham and Wolff [2]. They observed that the severity of migraine headaches lessened after the injection of ergotamine tartrate as the amplitude of temporal artery pulsation declined. Vascular theory proposed the serotonin released from the platelets in blood vessels to be the causal molecule. Because serotonin has constrictive effects on blood vessels, local cerebral blood flow is decreased, and this was thought to cause the auras that occur in migraines. After the reuptake or metabolization of this serotonin, the constricted blood vessels dilate, inducing the pain of the migraine headache.

The trigeminovascular theory of migraine pathophysiology was proposed by Moskowitz [3]. In this revolutionary theory, the trigeminal nerve is thought to be the primary origin of migraine headaches. Stimulation of the trigeminal nerve causes vascular dilatation, and neurogenic inflammation also contributes to migraine symptoms. However, the theory does not explain the cause of trigeminal nerve excitation.

## 3. Evidence against the Vascular and Trigeminovascular Theories of Migraine

In the vascular and trigeminovascular theories, vascular dilatation is believed to be the cause of migraine headaches. However, evaluations of arterial dilatation using cerebral catheter angiography and cerebral blood flow using nucleological examination during migraines in sufferers have found no intra- or extracranial arterial dilatation or increase in cerebral blood flow [9,10].

Many drugs and neurotransmitters have been found to induce migraine headaches. Sildenafil (Viagra^®^) is a selective inhibitor of cyclic guanosine monophosphate hydrolyzing phosphodiesterase. This increases cyclic guanosine monophosphate and induces migraine headaches [5]. Kruuse examined the cerebral blood flow velocity of the middle cerebral arteries (MCA) by transcranial Doppler ultrasonography and regional cerebral blood flow in the territory of the MCA using single photon emission computed tomography (SPECT) [11]. Sildenafil induced migraine attacks in 10 of 12 migraine without aura patients. However, both blood flow velocity and regional cerebral blood flow remain unchanged after sildenafil intake. This led to the suggestion that the site of action of sildenafil in the induction of migraine is not the blood vessel itself, but the perivascular sensory nerve terminals. The activation of adenosine triphosphate (ATP)-sensitive potassium (KATP) channels in arterial smooth muscles contributes to vasodilation, and the inhibition of KATP channels induces constriction of the middle meningeal arteries [12]. Migraine-triggering molecules such as levcromakalim activate and open KATP channels, triggering migraines [13]. Yet, measurement of MCA blood flow velocity during migraines found no change. Thus, levcromakalim does not act on KATP channels in human cerebral arteries, and the induction of migraine headaches by KATP channel openers cannot be explained by cerebral arterial dilatation.

A recent study used 3.0 Tesla magnetic resonance angiography (MRA) to evaluate human cerebral blood vessels. Schoonman et al. examined the MRA findings of cerebral blood vessels during nitroglycerine-induced migraine attacks and found no vasodilatation of cerebral or meningeal blood vessels [14]. Similarly, Amin et al. examined MRA data gathered during spontaneous unilateral migraine attacks in 19 female migraine patients and found no extracranial or meningeal arterial dilatation and only slight intracranial arterial dilatation. They speculated that the slight dilatation of the cerebral arteries might be a consequence of the pain rather than its cause [15].

Mirza et al. examined patients with throbbing dental pain, the rhythm of which appeared to correspond to the arterial pulse [16]. Simultaneous recording of the throbbing rhythm and arterial pulse demonstrated that the throbbing rate (which had a mean of 44 beats per minute (bpm)) was much slower than the arterial-pulsation rate (which had a mean of 73 bpm) (*p* < 0.001) and that the two rhythms exhibited no underlying synchrony. This demonstrated that the physiological mechanisms underlying these rhythmic events are distinct. We can infer from this that the pulsatile throbbing pain experienced by migraine sufferers may also not be reflective of arterial pulsation. However, a similar study for patients with migraine headaches has not been reported.

In trigeminovascular theory, a process called plasma protein extravasation (PPE) is thought to occur after neurogenic perivascular inflammation and arterial dilatation. Ultrasmall superparamagnetic iron oxide (USPIO)-enhanced magnetic resonance imaging (MRI) has confirmed the existence of macrophage-mediated inflammation. It has been suggested that the phosphodiesterase-3-inhibitor, cilostazol, could induce migraine headache in patients with migraine without aura; however, a study found no association between cilostazol and USPIO signals on the headache side, indicating that PPE is not a cause of migraines [17]. Cortical spreading depression (CSD) in a rat model has been shown to induce brief dilatation and prolonged constriction of pial arteries, prolonged dilatation of dural arteries, and PPE. Yet, despite the clinical efficacy of anti-CGRP monoclonal antibodies in the prevention of migraines, the anti-CGRP monoclonal antibody, fremanezumab, has no dilatory or constrictive effects on these arteries and does not cause PPE [18]. This suggests that these arterial events and PPE are not mediated by CGRP and are not causative factors in migraine headaches. A particular class of CGRP receptor antagonists called gepants has demonstrated clinical efficacy in the abortion and prevention of migraine headaches [19,20]. However, research has demonstrated that CGRP receptor antagonists do not directly induce vasoconstriction, indicating that the effects of CGRP receptor antagonists cannot be due to such direct vasoconstriction [5]. CGRP is also not involved in plasma extravasation in rats; thus, the related inflammation cannot be a trigger for migraine attacks [21].

Based on these findings, the present author believes that vascular dilation is not the primary cause of migraine headaches and that the trigeminovascular theory does not explain the pathophysiology of migraine. It is apparent that the dilatation of blood vessels is the result of neuronal inflammation, not the cause of migraine headaches. CGRP is considered the initial trigger of trigeminal nerve stimulation causing migraine headaches because CGRP receptor antagonists and monoclonal antibodies are clinically effective to abort and prevent migraine headaches.

## 4. The Mechanisms of Triptans and Ditans in the Abortion of Migraine Attacks

Triptans are a class of drugs used in the acute treatment of migraine. It has been postulated that the mechanism of triptans is the stimulation of 5-HT_1B_ receptors that results from the constriction of dilated dural and cranial arteries [22]. However, as discussed in the previous section, research has demonstrated that arterial dilatation is not the cause of migraine headaches. More recently, it has been proposed that triptans are 5-HT_1D_ receptor agonists. 5-HT_1D_ receptors are located in the Aδ-fibers of the trigeminal nerve. The stimulation of these receptors suppresses the nociceptive transmission of Aδ-fibers of the trigeminal nerve. The majority (90%) of CGRP neurons in human trigeminal ganglions express 5-HT_1D_ [6].

The ditan, lasmiditan, is a selective 5-HT_1F_ receptor agonist [7] used as an acute medication for migraine headaches in the United States and Japan. Receptors of 5-HT_1F_ are present on both C-fibers and Aδ-fibers of the trigeminal nerve but not within the blood vessels. This means that lasmiditan does not constrict blood vessels, yet is effective against migraine headaches. This is further evidence that migraine headaches are not caused by vasodilatation. Since 5-HT_1F_ receptor agonists only suppress the release of inflammatory neurotransmitters, including CGRP, from the C-fiber of the trigeminal nerve and the nociceptive transmission of Aδ-fibers of the trigeminal nerve, the effects of triptans and ditans are only fully explained within the trigeminal nerve.

## 5. New Theory of the Peripheral Pathophysiology of Migraines

Since the trigeminal nerve is nociceptive, its stimulation is sufficient to cause headaches. In the new theory proposed by the present author, the primary cause of migraines is the activation of the cervical complex of the trigeminal nucleus and trigeminal nerve. This activation of the trigeminal nerve results in neurogenic inflammation, including CGRP release from the C-fibers of the trigeminal ganglion and peripheral trigeminal nerve terminal [6]. CGRP receptors are located in larger neurons and thicker fibers, which correspond to the Aδ-fibers of the trigeminal nerve [6]. Aδ-fibers are myelinated nociceptive nerves; thus, stimulation of Aδ-fiber CGRP receptors causes sharp severe headaches (Figure 1). Trigeminal CGRP neurons co-occur alongside neurons that produce other transmitters, such as pituitary adenylate cyclase-activating polypeptides (PACAPs) and the neuropeptide, substance P. The satellite glial cells in the trigeminal ganglia that surround the neuronal cells can be labeled with CGRP receptor antibodies. CGRP has been shown to activate the release of inflammatory cytokines and nitrous oxide (NO) from ganglionic glial cells. These substances can, in turn, promote CGRP release, creating a positive feedback loop [6]. Given that the trigeminal ganglion is central to the trigeminal pain pathway, it seems reasonable to speculate that blocking CGRP transmission within the trigeminal ganglion could abort migraines or prevent debilitating symptoms. During the headache phase of migraines, persistent activation of the trigeminal ganglion and its local CGRP circuits amplifies the strength and duration of pain signals along central trigeminal afferent fibers to the central nervous system [4].

Immunostaining has found that only a very small proportion of trigeminal neurons express both CGRP and CGRP receptor proteins [23]. The expression of CGRP receptors in the trigeminal ganglion is likely to be involved in signaling mechanisms, including the positive feedback loops important for sensitization in facial pain and headaches. CGRP released from neurons stimulates the surrounding glial cells, glial cell ATP-gated purinergic P2Y receptors, and neuronal P2X3 receptors, as well as inducing the release of NO and cytokines. The feedback loop causes more CGRP synthesis and leads to further CGRP release by paracrine mechanisms, and possibly also by autoregulatory autocrine mechanisms. These intraganglionic signals serve to sustain peripheral sensitization in migraines. CGRP may act on neurons to initiate a cAMP-signaling cascade, which activates the P2X3 gene. CGRP can also have indirect effects through the activation of the brain-derived neurotrophic factor (BDNF) gene and the release of BDNF from satellite glia. This results in the upregulation of P2X3 expression in neurons. CGRP also has the potential to function as a paracrine factor in the stimulation of nearby glial cells and neurons, inducing the CGRP feedback loop described above. Direct autocrine regulation of the CGRP gene in the trigeminal ganglia has been demonstrated using primary cultures.

Mast cells are the major inflammatory regulators in the trigeminal ganglion and dura. CGRP stimulation of dural mast cells in rats has been shown to cause histamine release [24]. Histamine can then stimulate C-fibers to release CGRP and substance P. However, it may not be possible to extrapolate the role of mast cells in human migraines from these findings in rats because human dural mast cells do not have CGRP receptors [21].

The vascular and trigeminovascular theories of migraine suggest that vasodilatation is the cause of migraine headaches [2,3]. While inflammatory neurotransmitters released from the trigeminal nerve are potent vasodilators, trigeminal nerve stimulation and the onset of migraine headaches occur before these neurotransmitters affect the blood vessels and trigger vasodilatation. Capsaicin-induced extravasation is C-fiber-dependent, and substance P and neurokinin A, but not CGRP, cause dose-dependent extravasation in the dura [25]. Pial vessels are richly innervated by sensory fibers and demonstrate no neurogenically mediated extravasation. The tight endothelial junctions of the blood–brain barrier likely act as a physical barrier to extravasation from the pial circulation.

## 6. The Central Neuronal Theory of Migraine Pathophysiology

Leo and Morison first reported CSD in rabbits, although they described the phenomenon as spreading cortical depression rather than CSD [26]. The concept was not related to migraine pathophysiology until the neural theory proposed that CSD is the origin of migraines [1].

In 2001, Hadjikhani et al. utilized functional magnetic resonance imaging (fMRI) with three migraine patients while they were experiencing visual auras. They observed that the blood oxygenation level-dependent (BOLD) signal in these patients progressed slowly and contiguously over the occipital cortex in congruence with the visual percept. The BOLD signal is not directly related to blood flow but reflects the balance between oxygen delivery and oxygen consumption. Thus, the authors concluded that migraine auras are not evoked by ischemia but by the aberrant firing of neurons [27].

CSD is now known as the cause of migraine auras. Most migraine attacks do not precede aura symptoms, and a cerebral blood flow study found no blood flow changes during migraine attacks without auras but did find hypovolemia during migraines with aura [1]. Denuelle et al. investigated seven patients with migraine without aura using positron emission computed tomography (PET) and detected cerebral hypoperfusion during migraine attacks and posterior cortical hypoperfusion after the attacks [28]. They, therefore, suggested common pathogenesis for migraines with and without auras.

Analysis of in vitro brain slices and in vivo astrocytic cultures has shown that CSD induces reactive astrocytosis and elevated expression of proinflammatory markers such as cytokines, chemokines, and reactive oxygen intermediates but does not affect the number and volume of neurons [29]. The effective prevention of migraines by anticonvulsants such as topiramate and valproate was established through research that demonstrated a relationship between the frequency of evoked CSDs after topical potassium application as well as changes in the cathodal stimulation threshold required to evoke CSD in rats following anticonvulsant administration. Long-term daily administration of these prophylactic drugs suppresses CSD frequency by 40–80% in a dose-dependent manner and increases the cathodal stimulation threshold [30]. It is possible that CSD or CSD-like events may also play a role in the pathophysiology of migraines without auras because CSD is generated within clinically silent brain regions. Prophylactic migraine drugs, such as topiramate, valproate, propranolol, amitriptyline, and methysergide, have been implicated in the inhibition of glutamate release and the blocking of N-methyl-D-aspartate (NMDA) receptors. Because long-term treatment is required for these drugs to be effective, they may cause long-term modulation of the expression of genes or their encoded proteins. There is copious evidence that CSD is the cause of neuronal inflammation and the activation of the central and peripheral trigeminal nerves in migraines. CSD involves the local release of ATP, glutamate, potassium, and hydrogen ions by neurons, glia, and vascular cells and of CGRP and NO by activated perivascular nerves. CSD causes neuronal pannexin1 (Panx1) mega channels to open and activates the release of caspase-1 and high-mobility group box 1 (HMGB1) from neurons and nuclear factor kappa B (NFκB) activation in astrocytes [31]. These proinflammatory molecules diffuse toward the surface of the cortex, where they activate pial nociceptors, triggering neurogenic inflammation and persistent activation of dural nociceptors and the subsequent activation of central trigeminal neurons in the trigeminal spinal nucleus [32].

Glutamate receptor antagonists impair CSD propagation, and glial glutamate transporters (GLT) are responsible for clearing extracellular glutamate. A study found that GLT-1 knockout mice show increased CSD frequency and velocity compared to controls. These mice also exhibited a more rapid accumulation of glutamate in the extracellular space during the early phase of CSD than the control mice [33]. However, there are no data about GLT-1 in human migraine patients. Memantine is a glutamate NMDA receptor antagonist currently approved for the treatment of Alzheimer’s disease. Several studies have also shown memantine to be clinically effective in the treatment of migraines. In a 12-week randomized double-blind controlled trial, memantine produced a significant reduction in the monthly attack frequency [34]. Patients administered memantine had fewer work absence days, reduced migraine severity, and lower disability scores than patients in the placebo group in both intention-to-treat and complete case analyses. Thus, glutamate NMDA must play some role in the pathophysiology of migraines.

Cortical habituation and sensitization were compared between migraine patients and healthy controls using median somatosensory-evoked potentials (MSEPs) [35]. Three consecutive MSEPs, averaging 100 epochs each, were recorded. The N19 amplitude of block 1 was considered for sensitization, and the amplitudes of blocks 2 and 3 were compared with that of block 1 to measure habituation and augmentation. Among migraine sufferers, 71% showed a lack of habituation, compared to 27.6% of healthy controls. This lack of habituation was observed in all migraine subtypes. Augmentation was present in 61% of migraineurs, with greater frequency in chronic migraine sufferers (73.9%) than those with episodic migraines (63%) and medication overuse headaches (48%). Sensitization was not significantly different between migraineurs and controls, but patients with allodynia showed greater sensitization than those without. Habituation is a protective mechanism that prevents excessive repetitive discharge from neurons, thereby reducing neuronal stress and the accumulation of lactates and protons, which are responsible for CSD. It has been suggested that gamma-aminobutyric acid inhibitory neurons may be impaired in migraine sufferers. The cortical inter-neuronal inhibitory pathway may mature with age, explaining the dissipation of migraines in many patients after the fourth decade.

Silent ischemic lesions in migraineurs are often observed in the cerebellum [36]; however, the role of the cerebellum in migraines has not yet been fully elucidated. There is a higher prevalence of infratentorial hyperintensities on migraineur MRIs than in the general population. Recent work has revealed the cerebellum to be a sensory modulator, strategically localized to modify pain signaling in the brainstem. There is a rich supply of CGRP and its receptors in the cerebellum, in Purkinje cells, central nuclei, and the inferior olive, that support this modulation [21].

Ischemic stroke is strongly associated with migraines with auras [37]. There is also growing evidence of an association between migraines and other cardiovascular disorders, including myocardial infarction, hypertension, venous thromboembolism, and atrial fibrillation, with a stronger association in those with migraine with aura. The association between migraines and perioperative stroke is strongest in patients with migraine with aura and migraine sufferers without cardiovascular risk factors. Migraine is also a risk factor for silent brain lesions such as white matter MRI hyperintensities and infarct-like lesions. Traditional cardiovascular risk factors and migraine medications do not modify the association between structural brain lesions and migraines. White matter MRI hyperintensities are associated with an increased risk of stroke, dementia, and death. However, longitudinal studies found that migraineurs with these hyperintensities do not have an increased risk of cognitive decline. Unfortunately, the cross-sectional design of most of these studies prevents the inference of a cause–effect relationship between migraines and vascular pathologies.

Autoregulation dynamics in patients with migraine have been assessed using spontaneous blood pressure fluctuations, respiratory-induced blood pressure oscillations, and simultaneous transcranial Doppler monitoring. This revealed a reduction in autoregulation in the posterior inferior cerebellar artery and the MCA in those with migraine with aura, but not in those with migraine without aura [38].

## 7. Migraine Generation in the Central Nervous System

Although CSD appears to be the cause of migraine auras, there are premonitory symptoms that occur a few days before the aura, which cannot result from it. Thus, a full explanation of migraine pathophysiology requires other migraine generators in the central nervous system. High spatial and temporal resolution imaging allows for observation and evaluation of real-time dynamic changes in brain functions. PET studies have demonstrated that the left dorsal rostral pons is activated during spontaneous human migraine attacks. This activation persisted after the injection of sumatriptan despite complete cessation of the symptoms of headache and photo- and phonophobia [39,40]. Areas of the brainstem that are activated during migraines include the locus coeruleus and the dorsal raphe nucleus, both of which are involved in the anti-nociceptive network. The raphe and the dorsolateral pontine tegmentum are involved in sleep and arousal. Migraine sufferers often experience changes in arousal levels over the various phases of a migraine attack. Activation has also been observed in the right anterior cingulate, the posterior cingulate, the cerebellum, the thalamus, the insula, the prefrontal cortex, and the temporal lobes. Together, these areas are known as the central pain matrix. The anterior cingulate is the area that has shown the most consistent activation across various pain studies. It is thought to be involved in the affective and evaluative dimensions of pain. The insula has also shown consistent activation across pain studies. The insula has connections with the limbic system and autonomic nervous system and is thought to be involved in the emotional aspects of pain. The prefrontal cortex is involved in the cognitive–emotional processing of pain.

The high spatial and temporal resolution and non-invasive nature of MRI have made it a popular tool in brain research, including research into migraines [41]. Increased fMRI signal intensities in the red nucleus and substantia nigra have been observed before the onset of migraine symptoms. This is followed by signal changes in the occipital cortex. There is also migraine-specific brainstem activation and serotonergic and noradrenergic modulation of cortical activity and attentiveness to environmental stimuli [42]. This may help to explain the associated symptoms of migraine, such as photophobia and phonophobia. The right anterior cingulate is activated irrespective of the side on which the pain occurs, indicating the right side cerebral laterality of pain perception and control. Schulte et al. examined the fMRI of a single migraine patient every day for 30 days. The period studied included three complete untreated spontaneous migraine attacks [43]. They found that the hypothalamus was significantly more active in the 24 h preceding the onset of migraine headaches and showed the greatest functional connectivity with the spinal trigeminal nuclei. During the ictal state, the hypothalamus was functionally coupled with the dorsal rostral pons, an area previously found to be activated during migraines in a PET study [39]. A similar study gathered data from 27 spontaneous migraine attacks in seven patients. Functional activation of the hypothalamus was observed in the 48 h preceding the onset of migraine headaches [44]. Hence, hypothalamic activation seems to be responsive for the premonitory symptoms that occur a few days before migraine attacks. Ziegeler et al. examined the fMRI of 27 migraine patients before and 2 weeks after administration of 70 mg of the CGRP receptor antibody, erenumab. After erenumab treatment, there was reduced activation of many pain matrix areas including the thalamus, middle temporal gyrus, operculum, and cerebellum. Hypothalamic activation was also significantly reduced in those patients who responded to the erenumab treatment [45]. These results support the theory that the hypothalamus is a migraine generator. Edvinsson et al. argue that the clinical features of migraine point toward the involvement of the hypothalamus in the initiation of migraine attacks [21]. These features include yawning, tiredness, and mood changes, which suggest a link between the hypothalamus and premonitory symptoms. The circadian rhythmicity of migraines is further evidence of hypothalamic association, as it suggests the involvement of melatonin. Finally, the correlations found between migraine attacks and hormonal fluctuations, such as those related to the menstrual cycle, further support the role of the hypothalamus in determining the onset of attacks.

Hypothalamic neurons regulate the firing of preganglionic parasympathetic neurons in the superior salivatory nucleus (SSN) and sympathetic preganglionic neurons in the spinal intermediolateral nucleus [46]. The SSN can stimulate the release of acetylcholine, vasoactive intestinal polypeptides (VIPs), and NO from meningeal terminals of postganglionic parasympathetic neurons in the sphenopalatine ganglion (SPG). The activation of SSN neurons can modulate the activity of central trigeminal neurons in the spinal trigeminal nucleus. The enhanced cranial parasympathetic tone during migraine is evidenced by the symptoms of lacrimation and nasal congestion [46].

Infra-slow oscillatory activity is an indicator of gliotransmitter release and can be measured with fMRI. In migraine patients, increased infra-slow oscillatory activity has been observed in the brainstem and hypothalamus immediately prior to migraine attacks [47]. Researchers have investigated every voxel in the brain during different phases of migraine, but only the brainstem, hypothalamus, and thalamus displayed altered oscillatory activity directly before a migraine headache. It is possible that increases in infra-slow oscillatory power and regional homogeneity (ReHo) within the brainstem and hypothalamus during the phase preceding attack result from enhanced amplitude and synchronization of oscillatory gliotransmitter release. Increased ReHo is a useful measure of synchronization. This was found to occur in the same regions of periaqueductal gray matter (PAG), the hypothalamus, and the thalamus during the phase immediately before migraine attacks. Since these increases in ReHo did not occur in the dorsal pons or the medulla, it would appear that the PAG and the hypothalamus are critical to the onset of migraine.

A magnetoencephalographic study identified the differences between migraine patients, strict criteria TTH patients, and healthy controls [48]. In response to paired-pulse electrical stimulations, the first gating responses from the contralateral primary somatosensory cortex were significantly smaller in the migraine group compared to the TTH group and controls. The first response reflected the pre-activation excitability. This factor is therefore a potential marker for differentiation between TTT and migraines.

CGRP and PACAP are able to modulate nociceptive neuronal activity in several key trigeminal regions [49]. The CGRP receptor complex is found in the trigeminal cervical complex and the spinal trigeminal tract. The posterior and paraventricular hypothalamic regions are activated by both dural vascular stimulation and migraine attacks. The manipulation of local chemicals such as PACAP modulates the transmission of durovascular nociceptive trigeminal neurons. However, it is not yet clear which sites are relevant for the generation and maintenance of migraine attacks.

## 8. Newly Proposed Central Pathophysiology of Migraine

In the premonitory phase of migraines, hypothalamic activation can be inferred [21,46,50]. Alterations in appetite and food cravings suggest changes in the hypothalamic region [51]. These alterations induce the activation of the trigeminal nucleus caudalis in the brainstem and lead to trigeminal nervous activation (Figure 2). Because the trigeminal nerve is nociceptive, the activation of the trigeminal nerve causes nociceptive signal input to the brain.

Migraine is conceptualized as a disorder of sensory network gain and plasticity, starting at the hypothalamus and trigeminocervical complex, and including the trigeminal subnucleus caudalis [52]. Stimulation of the C2 peripheral nerve field is an effective method of neuromodulation for the treatment of headaches [53]. We employed intermittent electro-acupuncture as a less invasive alternative to electrode and device implantation and demonstrated its clinical efficacy against migraines [54]. To investigate the mechanism behind these therapies, resting-state fMRI was examined at baseline and after 3 months of C2 peripheral nerve field stimulation with electro-acupuncture [55]. The migraine without aura group showed a significant decrease in functional connectivity between the right hypothalamus and the left insula. We can therefore conclude that C2-stimulating neuromodulation improves migraine headaches by modifying pain-related functional connectivity in the pain matrix.

Photophobia is a frequent symptom of migraine. A novel retinothalamocortical pathway that carries the photic signal from melanopsinergic and non-melanopsinergic retinal ganglion cells to thalamic neurons has been observed [56]. Migraine modifies the activity of these neurons, and their axonal projections convey signals about headache and light to the multiple cortical areas involved in the generation of migraine symptoms, including photophobia.

Pre-treatment of rats with lamotrigine has been shown to suppress CSD elicited by occipital KCl application [57]. Valproate did not affect the triggering of CSD, although it inhibited CSD propagation between the occipital and frontal electrodes. Valproate is generally effective in preventing migraine attacks. However, lamotrigine shows selective inhibition of migraine auras. Because lamotrigine is not effective for all migraine attacks, CSD may not be a critical component of all migraine headaches.

We used diffusion tensor imaging to compare migraine with aura and migraine without aura, patients overusing headache medications, and healthy controls [58]. The migraine without aura with medication overuse and the migraine with aura without medication overuse groups showed significant decreases in fractional anisotropy in several areas of white matter compared with the control group. The migraine without aura without medication overuse headaches group was not significantly different from the control group. The sample size of migraine with aura patients who were overusing headache medication was too small for statistical analysis. Nevertheless, it was concluded that migraines with auras and medication overuse headaches include significant structural pathology of the white matter, and this pathology is more severe than that in migraine without aura patients. Neither disease duration nor the frequency of migraine attacks was correlated with the fractional anisotropy values of the corpus callosum. This suggests that this pathology may be the cause rather than the result of migraine headaches.

Premonitory symptoms and aura do not occur in every migraine attack. However, it is reasonable to suppose that there are functional changes in the thalamus, hypothalamus, and occipital cortex before a migraine attack, despite the absence of corresponding symptoms. In the pathophysiology of migraine, these functional changes provide a likely source for the stimulation of the trigeminal nucleus. However, the neural theory citing CSD as the origin of migraine attacks is not fully supported by current evidence. The present author proposes that migraine pathophysiology starts with functional changes in the hypothalamus, which may induce premonitory symptoms. In some patients, CSD occurs directly before migraine attacks. These functional changes in the hypothalamus and the CSD activate the trigeminal nucleus in the brainstem. The areas of brainstem activation during migraine attacks observed in PET and fMRI studies are the trigeminal nucleus, the locus coeruleus, the dorsal raphe nucleus, and the PAG.

## 9. Proposed Diagnostic Criteria for Headaches

Intravenous infusion of CGRP or PACAP induces migraine attacks in approximately 60%, not all of migraineurs [59]. Glutamate, CGRP, and nerve growth factor (NGF) are increased in chronic migraine patients [60]. NGF can upregulate CGRP expression in sensory and motor neurons. Increased CSF concentrations of NGF, BDNF, and glutamate have been found in fibromyalgia patients.

Currently, the diagnosis of headaches is based on clinical symptoms only, as there are no pathophysiological diagnostic tests available [8].

The present author would like to propose molecular-based diagnostic criteria for headaches. If CGRP is the major player in a headache, this headache should be diagnosed as a CGRP headache. For CGRP headaches, anti-CGRP medication, such as gepants and anti-CGRP-(receptor) monoclonal antibodies, should be effective. Patients with a clinical diagnosis of migraine in whom these anti-CGRP medications are ineffective should not be diagnosed with CGRP headaches. Other neurotransmitters, such as VIPs, amylin, and PACAP, are known to cause headaches.

The plasma half-life of CGRP following its infusion into humans is approximately 7 min during the fast decay phase and 26 min during the slower phase of decay [6]. This makes real-time measurement of CGRP levels in peripheral blood samples very difficult. In migraine, CGRP is released from the trigeminal nerve and then flushed out in the venous blood. As a result, the CGRP concentrations found in peripheral blood samples are not representative of the concentrations in or around the trigeminal nerve. High levels of CGRP release into the external jugular vein are not associated with increased plasma levels of CGRP in the cubital vein [4].

CGRP levels are highest in younger people and decline with age. In blood samples taken from the external jugular veins of patients, CGRP levels were found to be elevated during headaches, whereas the levels of other neuropeptides associated with the trigeminal and autonomic nerves remained unchanged. Elevated blood levels of CGRP were also observed during attacks of cluster headaches and chronic paroxysmal headaches. In contrast with migraines, however, these two conditions were associated with the release of VIPs. The instability and short half-life of these peptides limit reliable measurements.

CGRP is not the only neurotransmitter to induce pain in the trigeminal nerve. The CGRP family includes CGRP, adrenomedullin, amylin, and calcitonin. Adrenomedullin activates the CGRP receptor and therefore mimics the endogenous activity of CGRP [61]. CGRP and amylin share 50% of their sequence identity, with a high proportion of amino acids being identical at the N- and C- termini [61]. The amylin-1 receptor can also be activated by CGRP. Amylin antibodies frequently display cross-reactivity with CGRP. The major biological function of calcitonin is calcium homeostasis via osteoclast-mediated reabsorption and kidney excretion. Calcitonin is a potent and relatively long-acting agonist of both calcitonin and amylin receptors. Elevated levels of circulating pro-calcitonin have been reported in chronic migraine sufferers, with a significant increase in pro-calcitonin serum levels during migraine attacks compared with levels in the periods between attacks. The analgesic properties of calcitonin make it a promising treatment for migraine and other pain disorders. The anti-nociceptive properties of these peptides have been repeatedly demonstrated, although the exact mechanism of action is not yet fully understood. Amylin can have both pro-nociceptive and anti-nociceptive effects. Patients suffering from chronic migraines and, to a lesser extent, episodic migraines, display elevated concentrations of circulating amylin, as measured by enzyme-linked immunosorbent assays, relative to non-migraineurs. Recent evidence suggests that amylin plays a direct role in triggering migraine attacks, as infusion of the amylin mimetic, pramlintide, into migraine patients induced migraine-like attacks.

Procedures for the measurements of these neurotransmitters are yet to be clinically established due to their localized action, as the measurement of blood, plasma, or serum levels would not indicate changes in neurotransmitter expression in peripheral venous blood. The short half-life of these neurotransmitters also makes timely measurements difficult. It is hoped that, in the future, rapid and precise measures of these neurotransmitters will become available, allowing for the clinical application of these newly proposed diagnostic criteria. In vivo imaging such as optical imaging, ultrasonography, MRI, and nuclear medicine could determine the precise location and concentration of these neurotransmitters in the near future [62].

## Figures and Tables

**Figure 1 ijms-23-13002-f001:**
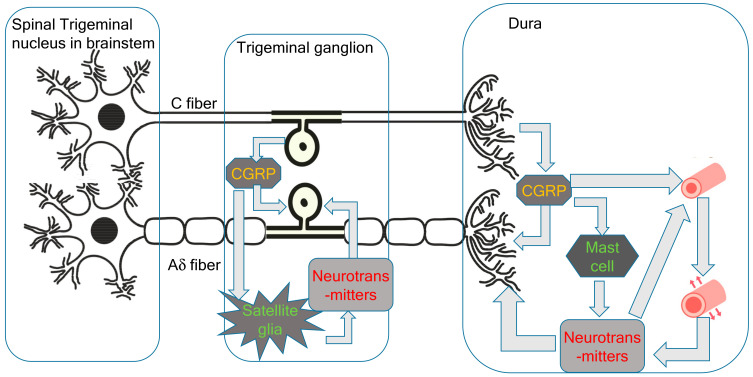
Peripheral pathophysiology of migraines.

**Figure 2 ijms-23-13002-f002:**
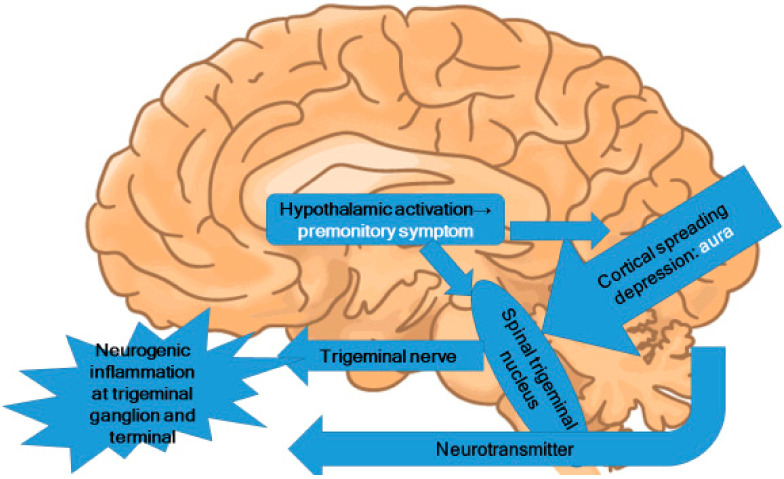
Central pathophysiology of migraines.

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
