# Peer review of "Migraine Pathophysiology Revisited: Proposal of a New Molecular Theory of Migraine Pathophysiology and Headache Diagnostic Criteria"

_ijms, 2022, doi:10.3390/ijms232113002_

Round 1
Reviewer 1 Report
The article presented for review tries to identify the new molecular theory of migraine pathophysiology and, at the same time, headache diagnostic criteria. The first sub-sections of the article indicate the existing knowledge. At this point, the author cites numerous items of literature, showing general knowledge on the subject matter. In the second part, quite similarly edited, the author presents a new theory of the peripheral pathophysiology of migraines. The author, in numerous citations and his assessment, very often indicates the dependence on calcitonin-gene-related peptides (CGRPs) After reviewing the presented article, I propose to accept it for publication.
Author Response
Thank you for your positive comments.
Yasushi Shibata
Reviewer 2 Report
The manuscript is very interesting and addresses an important question; what are the molecular pathomechanisms of migraine? The author discusses the existing theories of migraine pathophysiology and introduces a new molecular theory for this and for headache diagnosis. The new theory is based on MRI and PET measurements and brings new insights into the pathophysiology of headache.
The author could determine the major brain areas involved in the development of migraine. In addition, PET measurements were performed during different treatments to determine their mechanism of action.
While the paper is well written and discusses the pathomechanisms of headache in great details, the following statements on which the new theory is based should be accompanied by more evidences from the literature. Otherwise, the statements are merely speculative
- The pulsatile, throbbing pain experienced by migraine patients is not due to of arterial pulsation. This statement has never been supported with experimental data.
- It is clear that CGRP antagonists have no vasoconstrictor effect, but they are effective against migraine. Thus, this fact is not evidence for the statement that the vasodilatation is not a cause of migraine
The final conclusion is not sufficiently clear to me. The authors concluded that CGRP is the most important neurotransmitter for headache classification. However, its determination from the blood is challenging due to many technical difficulties. I would expect the conclusion here to suggest several alternative measurements that could help determine the new diagnostic markers for migraine.
Author Response
Thank you for your valuable comments.
1. The pulsatile, throbbing pain experienced by migraine patients is not due to of arterial pulsation. This statement has never been supported with experimental data.
-> I agree with you. So I changed the description as follow.
We can infer from this that the pulsatile throbbing pain experienced by migraine sufferers may also not be reflective of arterial pulsation. However, a similar study for patients with migraine headaches has not been reported.
2. It is clear that CGRP antagonists have no vasoconstrictor effect, but they are effective against migraine. Thus, this fact is not evidence for the statement that the vasodilatation is not a cause of migraine
-> In trigeminovascular theory, arterial dilatation, neurogenic perivascular inflammation, and plasma protein extravasation are thought to occur. As you pointed out, CGRP antagonists or monoclonal antibodies do not involve these events, however, are effective for migraine abortion and prevention. I would like to explain that CGRP-induced migraine headache does not involve arterial dilatation, neurogenic perivascular inflammation, and plasma protein extravasation at the beginning of the headache phase. So I added the following statement.
CGRP is considered the initial trigger of trigeminal nerve stimulation causing migraine headaches because CGRP receptor antagonists and monoclonal antibodies are clinically effective to abort and prevent migraine headaches.
3. The final conclusion is not sufficiently clear to me. The authors concluded that CGRP is the most important neurotransmitter for headache classification. However, its determination from the blood is challenging due to many technical difficulties. I would expect the conclusion here to suggest several alternative measurements that could help determine the new diagnostic markers for migraine.
-> Thank you for your comments. I added the following statement and one reference.
In vivo imaging such as optical imaging, ultrasonography, MRI, and nuclear medicine could determine the precise location and concentration of these neurotransmitters in near future [62].
All changes were highlighted.
Thank you for your comments and review to my manuscript.
Yasushi Shibata